

# Freshwater gastropods diversity hotspots: three new species from the Uruguay River (South America)

Diego E. Gutiérrez Gregoric[1,2] and Micaela de Lucía[2]

[1] Centro Científico Tecnológico La Plata, Consejo Nacional de Investigaciones Científicas y Técnicas, La Plata, Buenos Aires, Argentina
[2] División Zoología Invertebrados, Museo de La Plata, Facultad de Ciencias Naturales y Museo, Universidad Nacional de La Plata, La Plata, Buenos Aires, Argentina

Corresponding author
Diego E. Gutiérrez Gregoric,
dieguty@fcnym.unlp.edu.ar

## ABSTRACT

**Background:** The Atlantic Forest is globally one of the priority ecoregions for biodiversity conservation. In Argentina, it is represented by the Paranense Forest, which covers a vast area of Misiones Province between the Paraná and Uruguay rivers. The Uruguay River is a global hotspot of freshwater gastropod diversity, here mainly represented by Tateidae (genus *Potamolithus*) and to a lesser extent Chilinidae. The family Chilinidae (Gastropoda, Hygrophila) includes 21 species currently recorded in Argentina, and three species in the Uruguay River. The species of Chilinidae occur in quite different types of habitats, but generally in clean oxygenated water recording variable temperature ranges. Highly oxygenated freshwater environments (waterfalls and rapids) are the most vulnerable continental environments. We provide here novel information on three new species of Chilinidae from environments containing waterfalls and rapids in the Uruguay River malacological province of Argentina.

**Materials and Methods:** The specimens were collected in 2010. We analyzed shell, radula, and nervous and reproductive systems, and determined the molecular genetics. The genetic distance was calculated for two mitochondrial markers (cytochrome *c* oxidase subunit I–COI- and cytochrome b -Cyt b-) for these three new species and the species recorded from the Misionerean, Uruguay River and Lower Paraná-Río de la Plata malacological provinces. In addition, the COI data were analyzed phylogenetically by the neighbor-joining and Bayesian inference techniques.

**Results:** The species described here are different in terms of shell, radula and nervous and reproductive systems, mostly based on the sculpture of the penis sheath. Phylogenetic analyses grouped the three new species with those present in the Lower Paraná-Río de la Plata and Uruguay River malacological provinces.

**Discussion:** Phylogenetic analyses confirm the separation between the Uruguay River and the Misionerean malacological provinces in northeast Argentina. These new endemic species from the Uruguay River add further support to the suggestion that this river is a diversity hotspot of freshwater gastropods (with 54 species present in this basin, 15 of them endemic). These endemic species from environments with rapids and waterfalls should be taken into account by government agencies before the construction of dams that modify those ecologic niches in the Uruguay River.

## INTRODUCTION

Highly oxygenated freshwater environments (waterfalls and rapids) are the most vulnerable continental environments globally, supporting highly specific faunas (including gastropods) with narrow habitat requirements. Accordingly, many native snail populations are declining in numbers as a consequence of the continuous degradation and destruction of their natural ecosystems from unabated human activity (*Rumi et al., 2006*; *Strong et al., 2008*; *Darrigran & Damborenea, 2011*). In particular, freshwater gastropods (approximately 5% of the world's gastropod fauna) are at a disproportionately high risk of extinction (*Strong et al., 2008*). Of the 310 mollusc species listed as extinct in the 2015 International-Union-for-the-Conservation-of-Nature (IUCN) Red List of Threatened Species (http://www.iucnredlist.org), 73 (c. 23%) are gastropods from inland waters. The changes that result from damming rivers with waterfalls and rapids have caused the extinction of species—for example, those of the gastropod genus *Aylacostoma* (*Mansur, 2000a*; *Mansur, 2000b*). Despite the significance of this type of environment, the study of freshwater gastropods inhabiting waterfalls and rapids is poor (*e.g. Ponder, 1982*; *Glöer, Albrecht & Wilke, 2007*; *Gutiérrez Gregoric, Núñez & Rumi, 2010*). *Vogler et al. (2014)* described a new species of *Aylacostoma* from rapids in the High Paraná River (Argentina-Paraguay), based on materials collected in 2007. In 2011, however, the locations were flooded during the last stage of filling the Yacyretá Reservoir.

The Atlantic Forest—in Argentina represented by the Paranense Forest, occupying a large part of Misiones Province—constitutes one of the global priority ecoregions for biodiversity conservation. The orography of Misiones Province is rather accentuated and marked by a central ridge that acts as a watershed between the two great international rivers, the Paraná and the Uruguay—respectively of the Misionerean and Uruguay River malacological provinces as defined by *Núñez, Gutiérrez Gregoric & Rumi (2010)*. The Uruguay River is among the global hotspots of freshwater gastropod diversity according to *Strong et al. (2008)*, within the category of "Large rivers and their first and second order tributaries." This hotspot is represented mainly by the Tateidae (genus *Potamolithus*, with 12 endemic species), and to a lesser extent by Chilinidae (three endemic species) and Ampullariidae (endemic genus *Felipponea* with three species). The streams of Misiones Province contain waterfalls and rapids that have been poorly studied by malacologists. In these environments several endemic freshwater gastropod entities have been recorded—*e.g.,* the genera *Acrorbis* (Planorbidae), inhabiting only waterfall environments (*Hylton Scott, 1958*; *Ituarte, 1998*; *Rumi et al., 2006*) and *Felipponea* spp. (Ampullariidae), recorded in the rapids of the Uruguay River and its tributaries (*Castellanos & Fernández, 1976*; *Rumi et al., 2006*) and the species *Chilina megastoma* Hylton Scott, 1958 (Chilinidae), inhabiting the waterfalls of Iguazú National Park (Argentina and Brazil) (*Hylton Scott, 1958*; *Ituarte, 1997*), *Chilina iguazuensis* Gutiérrez Gregoric & Rumi, 2008 (Chilinidae) and

*Sineancylus rosanae* (Gutiérrez Gregoric, 2012) (Planorbidae), with the last being present in the rapids of the upper Iguazú River (Argentina and Brazil) (*Gutiérrez Gregoric & Rumi, 2008*; *Gutiérrez Gregoric, 2012*; *Gutiérrez Gregoric, 2014*).

The family Chilinidae (Gastropoda, Hygrophila) is one of the oldest families of freshwater gastropods (*Duncan, 1960*). Of the 21 species of the *Chilina* genus found in Argentina, 15 are endemic and nine of this 21 are vulnerable (*Rumi et al., 2006*; *Núñez, Gutiérrez Gregoric & Rumi, 2010*). Vulnerability was assessed based on one or more of the following: (1) known only from the type locality (three species); (2) no recent record (four species); (3) continuous restricted distribution (six species); (4) discontinuous restricted distribution (three species) (*Rumi et al., 2006*; *Gutiérrez Gregoric & Rumi, 2008*; *Gutiérrez Gregoric, Ciocco & Rumi, 2014*). Of those nine vulnerable species, four are in protected areas. Globally, the IUCN Red List of Threatened Species (http://www.iucnredlist.org) lists only one species as "vulnerable" (*C. angusta* (Philippi, 1860) from Chile), seven as "data-deficient," and four as "least concern." In the Del Plata basin (containing the Paraná, Uruguay and Río de la Plata rivers) six species of Chilinidae have been recorded. Three are found in the Lower Paraná-Río de la Plata and the Uruguay River malacological provinces: *Chilina fluminea* (Maton, 1809), *Chilina rushii* Pilsbry, 1896 and *Chilina gallardoi* Castellanos & Gaillard, 1981 (*Núñez, Gutiérrez Gregoric & Rumi, 2010*). The other three are from the Misionerean malacological province: *Chilina guaraniana* Castellanos & Miquel, 1980, originally recorded in the Paraná River in the area of the current Yacyretá reservoir but not having been cited since 1935, and the aforementioned *C. megastoma* and *C. iguazuensis* both recorded only in the Iguazú River and its tributaries (Argentina-Brazil) (*Castellanos & Gaillard, 1981*; *Gutiérrez Gregoric, 2008*; *Gutiérrez Gregoric, 2010*; *Gutiérrez Gregoric & Rumi, 2008*; *Núñez, Gutiérrez Gregoric & Rumi, 2010*).

In this study we describe and provide information on the anatomy and molecular genetics of three news species: *Chilina nicolasi*, *Chilina santiagoi* and *Chilina luciae* from rapids and waterfalls of the Uruguay River malacological province. Phylogenetic analyses were used to confirm the segregation of the three species and of the species in the different freshwater malacological provinces of Argentina.

## MATERIALS AND METHODS

The specimens were collected in the Misiones Province (authorized by the Ministry of Ecology, Natural Renewable Resources and Tourism) and deposited in the Malacological Collection at the Museo de La Plata, Buenos Aires Province, Argentina (MLP-Ma). Additional material in MLP-Ma was also studied. Adult specimens were first relaxed in menthol for 12 h, then immersed in hot water (70 °C), and finally stored in 96% (v/v) aqueous ethanol or fixed in modified Raillet-Henry (R-H) solution for freshwater animals—93% (v/v) distilled water, 2% (v/v) glacial acetic acid, 5% (v/v) formaldehyde, and 6 g sodium chloride per liter. Six shell measurements were taken: total length (TL), length of the last whorl (LWL), aperture length (AL), total width (TW), aperture width (AW), and aperture projection (AP) following *Martín (2003*; Fig. 1). For anatomical studies of the reproductive and pallial systems, the methodology of *Cuezzo (1997)* was followed.

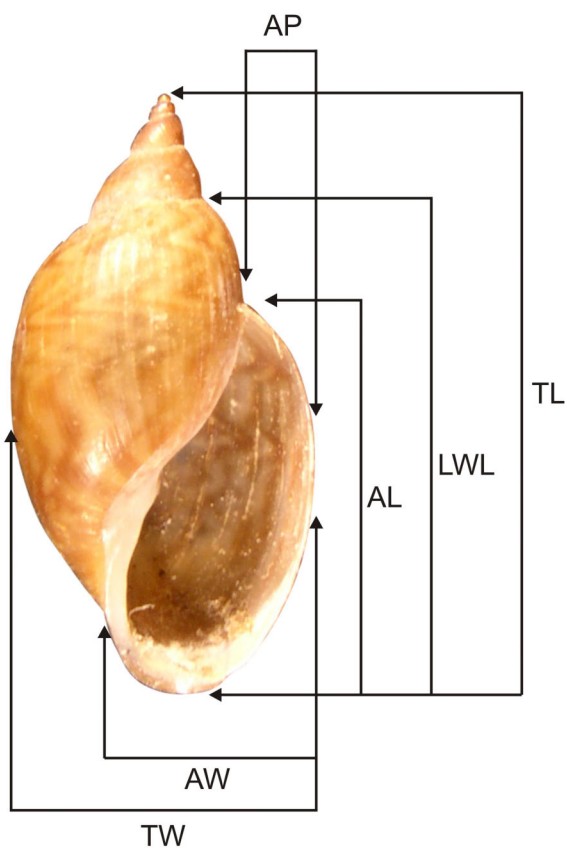

**Figure 1 Shell measurements used for Chilinidae.** TL, Total length; LWL, last whorl length; AL, aperture length; TW, total width; AW, aperture width; AP, aperture projection.

Dissections were done under a Leica MZ6 stereoscopic microscope and anatomical systems drawn with the help of a *camera lucida*. Figures were drawn only for characters that showed specific differences. The terminology used for the anatomical descriptions follows *Ovando & Gutiérrez Gregoric (2012)*. In addition, we compared these new species with species of Chilinidae for which anatomical and conchological studies have been undertaken: *Chilina megastoma* studied by *Ituarte (1997)* and *C. iguazuensis* described by *Gutiérrez Gregoric & Rumi (2008)* from Misionerean malacological province, Argentina; *C. fluminea fluminea* and *C. fluminea parva* Martens, 1868 studied by *Lanzer (1997)* from Río Grande do Sul, Brazil: *C. fluminea fluminea* studied by *Gutiérrez Gregoric (2008)* from Lower Paraná–Río de la Plata malacological province, Argentina; *C. rushii* and *C. gallardoi* studied by *Gutiérrez Gregoric (2010)* from Uruguay river and Lower Paraná–Río de la Plata malacological provinces, Argentina; *C. lilloi* Ovando & Gutiérrez Gregoric, 2012, *C. portillensis* Hidalgo, 1880 and *C. tucumanensis* Castellanos & Miquel, 1980 all studied by *Ovando & Gutiérrez Gregoric (2012)* from Middle Paraná and Central malacological provinces, Argentina; *C. mendozana* Strobel, 1874, *C. parchappii* (d'Orbigny, 1835), *C. cuyana* Gutiérrez Gregoric, Ciocco & Rumi, 2014 and *C. sanjuanina* Gutiérrez Gregoric, Ciocco & Rumi, 2014 all studied by *Gutiérrez Gregoric, Ciocco & Rumi (2014)* from Cuyo malacological province, Argentina.

The radulae were separated from the buccal mass and cleaned following the method of *Holznagel (1998)*, and mounted for scanning electron microscopy. The radular-dentition formula used is L–C–L (number lateral teeth–central tooth–number lateral teeth; there is no distinction between marginal and lateral teeth as there is in other molluscs).

Total DNA was extracted from c. 2 mm$^3$ samples from the foot of recently collected specimens (2010) using commercial kits (DNeasy Blood & Tissue, for Qiagen). A partial sequence of the genes encoding the mitochondrial cytochrome *c* oxidase subunit I (COI) and cytochrome b (Cyt b) were amplified by the polymerase chain reaction (PCR) with the universal primers of *Folmer et al. (1994)* and *Merritt et al. (1998)* respectively. Amplification was performed in a final volume of 50 µl, following *Gutiérrez Gregoric et al. (2013)* and *Gutiérrez Gregoric, Ciocco & Rumi (2014)*. The PCR products were purified with an AxyPrep PCR Clean-up Kit (Axygen Biosciences, Union City, CA, USA) and both DNA strands for each gene were then directly cycle-sequenced (Macrogen Inc., Seoul, South Korea). The resulting sequences were trimmed to remove the primers, and the consensus sequences of the individuals were compared to reference sequences in GenBank. Sequences of *C. megastoma*, *C. iguazuensis* and *C. fluminea* (partial) were obtained from the Barcode of Life Database (BOLD). The sequence alignment was performed with the Clustal X 2.0.12 software (*Larkin et al., 2007*), optimized by visual inspection and edited with a word processor. Since we obtained Cyt b sequences for only four individuals we calculated a pairwise genetic divergence (Kimura two-parameter) for this region, and only COI data were subjected to phylogenetic analyses by the methods of neighbor-joining (NJ) and Bayesian inference (BI). The NJ analysis was conducted using MEGA 5.05 software (*Tamura et al., 2011*) through the use of the maximum-composite-likelihood option for computing evolutionary distances (*Tamura, Nei & Kumar, 2004*). Statistical support for the resulting phylogeny was assessed by bootstrapping with 1,000 replicates (*Felsenstein, 1985*). The BI was carried out with the MrBayes 3.2 software (*Ronquist et al., 2012*). Two runs were performed simultaneously with four Markov chains that went for 1,000,000 generations, sampling every 100 generations. The first 10,000 generations of each run were discarded as burn-in, and the remaining 18,000 trees were used to estimate posterior probabilities.

The electronic version of this article in Portable Document Format (PDF) will represent a published work according to the International Commission on Zoological Nomenclature (*ICZN, 2012*), and hence the new names contained in the electronic version are effectively published under the *International Code of Zoological Nomenclature* from the electronic edition alone. This published work and the nomenclatural acts it contains have been registered in ZooBank, the online registration system for the ICZN. The ZooBank LSIDs (Life Science Identifiers) can be resolved and the associated information viewed through any standard web browser by appending the LSID to the prefix http://zoobank.org/. The LSID for this publication is: urn:lsid:zoobank.org:pub:3140E36D-B1F5-4C1B-9F3C-0081CDE88B00. The online version of this work is archived and available from the following digital repositories: PeerJ, PubMed Central and CLOCKSS.

## RESULTS

### Systematic account

**Family Chilinidae** *Dall, 1870*
**Genus** *Chilina Gray, 1828*

**Type species:** *Auricula* (*Chilina*) *fluctuosa Gray, 1828* (**subsequent designation of** *Gray, 1847*).

   **Diagnosis:** Species in the genus and family have an oval (oblong to ventricose) shell with an expanded last whorl. Nervous system with partial detorsion. Roof of the mantle cavity pigmented with kidney occupying almost entire length. Kidney inner wall with numerous transverse trabeculae of irregular contour. Rectum on right side of mantle cavity, anus near pneumostome. Incomplete division of male and female ducts; common duct opens to hermaphrodite duct, with irregular contours on both sides. Proximal portion of uterus with glandular walls. Calcareous granules in vaginal lumen and secondary bursa copulatrix or accessory seminal receptacle present. Penial terminal portion with cuticularized teeth-like structures.

   **Remarks:** The Chilinidae includes only the genus *Chilina* with 36 species, 21 of which are found in Argentina (*Núñez, Gutiérrez Gregoric & Rumi, 2010*; *Ovando & Gutiérrez Gregoric, 2012*; *Gutiérrez Gregoric, Ciocco & Rumi, 2014*) with the remainder in Chile and Brazil (*Castellanos & Gaillard, 1981*; *Simone, 2006*; *Valdovinos Zarges, 2006*).

### *Chilina nicolasi sp. nov.*

Urn:lsid:zoobank.org:act:A7D18E3D-1CA1-470F-A5B6-3EB070C994C3 (Figs. 2A, 2B, 3 and 4A–4E).

   **Type locality and type material:** Uruguay River, Alba Posse, Misiones Province, Argentina, (27°33′S; 54°40′W), coll. D.E. Gutiérrez Gregoric, V. Núñez & R.E. Vogler, 23 March 2010.

   Holotype: MLP-Ma 13412-2 (foot in alcohol, body in R-H, shell dry); paratypes: MLP-Ma 13412 same data (4 specimens: foot in alcohol, body in R-H, shell dry); MLP-Ma 14134 same data (10 specimens: body in R-H, shell dry).

   **Etymology:** Dedicated to the first son, Nicolás, of the first author of this paper.

   **Diagnosis:** Shell thick, oval, two columellar teeth (upper underdeveloped); radula with first lateral tooth with saw-like external side of mesocone; penis sheath twice the length of the prepuce; penis sheath inner sculpture with triangular regular pustules.

   **Description:**

   **Shell** (Figs. 2A and 2B). Thick, oval, periostracum light brown with weak dark reddish zigzag bands. Spire immersed. Last whorl well developed. Aperture 90% of LWL, slightly expanded, with white callus of terminal portion slightly widened and flattened. Width 73% of LWL. Aperture projected 35% of TW. Two columellar teeth, lower tooth more prominent and developed than upper. Dimensions: see Table 1.

   **Reproductive System** (Fig. 3). (i) Female reproductive system. Bursa copulatrix duct long (n = 2; 7.0 mm, 7.3 mm), five times bursa sac diameter. Bursa copulatrix sac

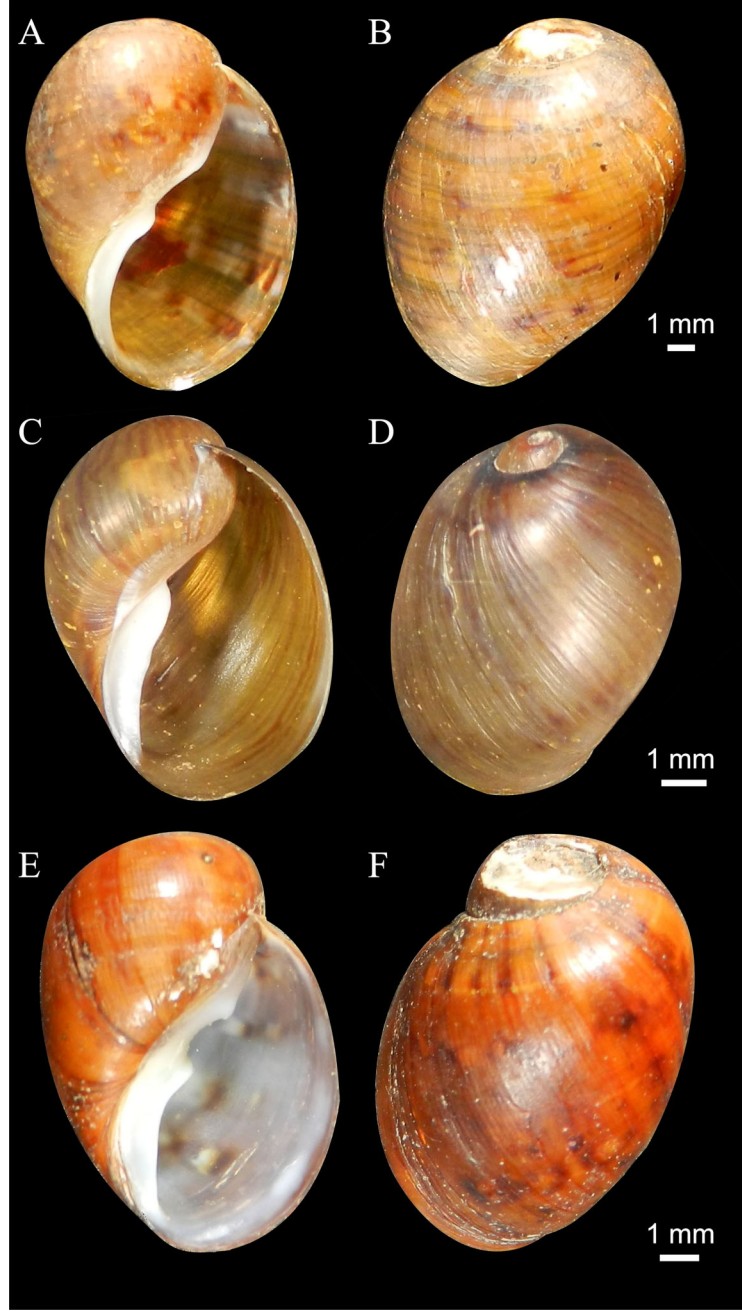

**Figure 2 Shells of new species (Holotypes).** (A–B) *Chilina nicolasi.* (C–D) *Chilina santiagoi.* (E–F) *Chilina luciae.*

spherical, located on left side of cephalopedal haemocoel between pericardial cavity and columellar base. Secondary bursa copulatrix short, emerging from base of uterus, cylindrical (*c.* 8% the length of bursa copulatrix duct). Vagina cylindrical, longer than wide, folded over free oviduct and entering female atrium. (ii) Male reproductive system. Prostate gland extending to lower half of uterus and consisting of variable size and with cylindrical acini. Vas deferens coiled twice, overlapping vagina. At level of penis complex,

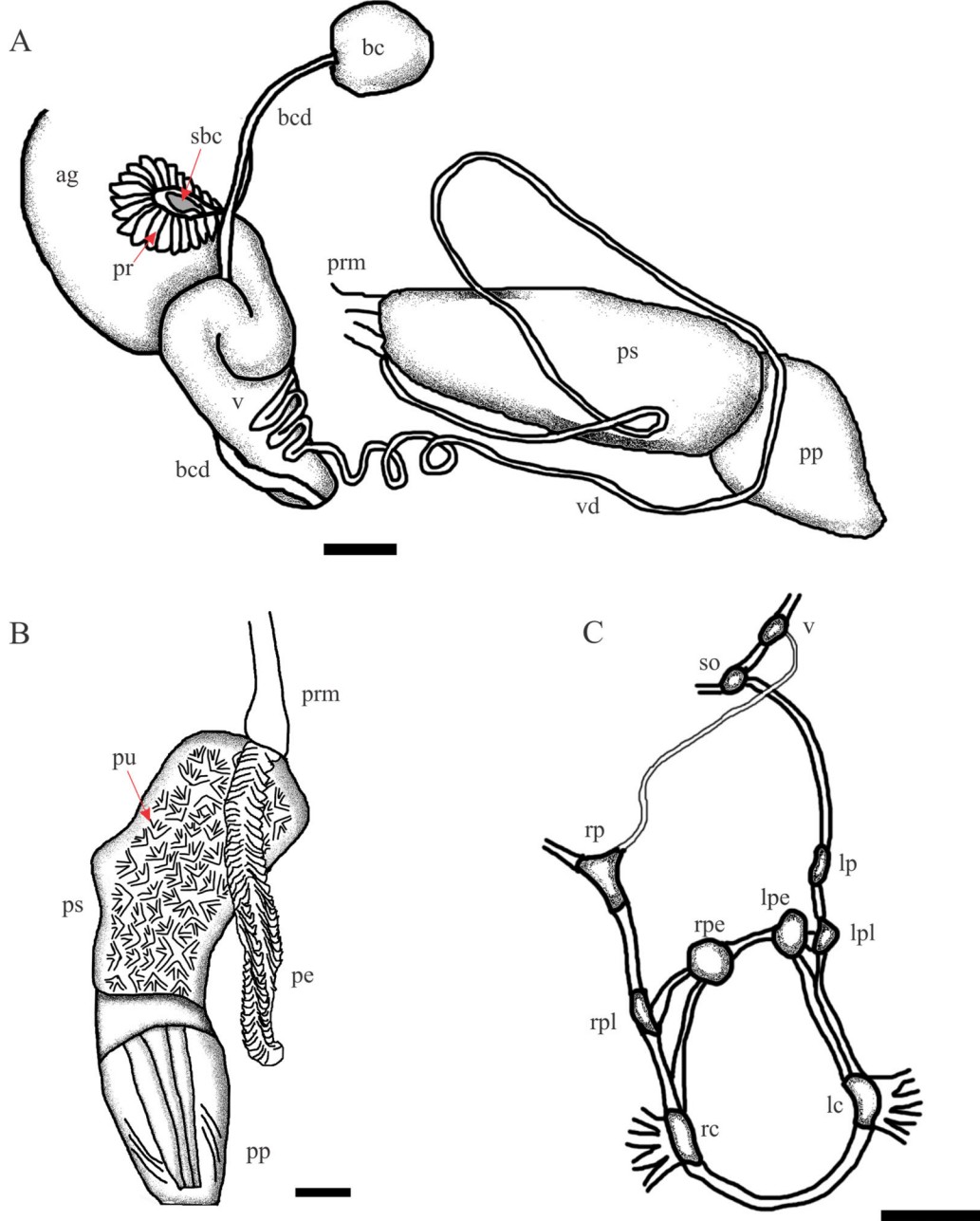

**Figure 3 *Chilina nicolasi* sp. nov.** (A) Diagram of dorsal view of part of the reproductive system. (B) Penis inner wall. Abbreviations: ag, albumen gland; bc, bursa copulatrix; bcd, bursa copulatrix duct; pe, penis; pr, prostate; pp, preputium; prm, penis retractor muscle; ps, penis sheath; pu, pustules; sbc, secondary bursa copulatrix; v, vagina; vd, vas deferens. (C) Diagram of nervous system. Abbreviations: lc, left cerebral; lpe, left pedal; lp, left parietal; lpl, left pleural; rc, right cerebral; rpe, right pedal; rp, right parietal; rpl, right pleural; so, subesphageal; v, visceral. Scale bar: 1.0 mm.

vas deferens bent back on itself. Penis sheath muscular, twice the length of the prepuce, with slight convexity on right side. Penis sheath inner sculpture with triangular pustules over entire surface. Penis elongated (as long as the penis sheath), robust, with outer

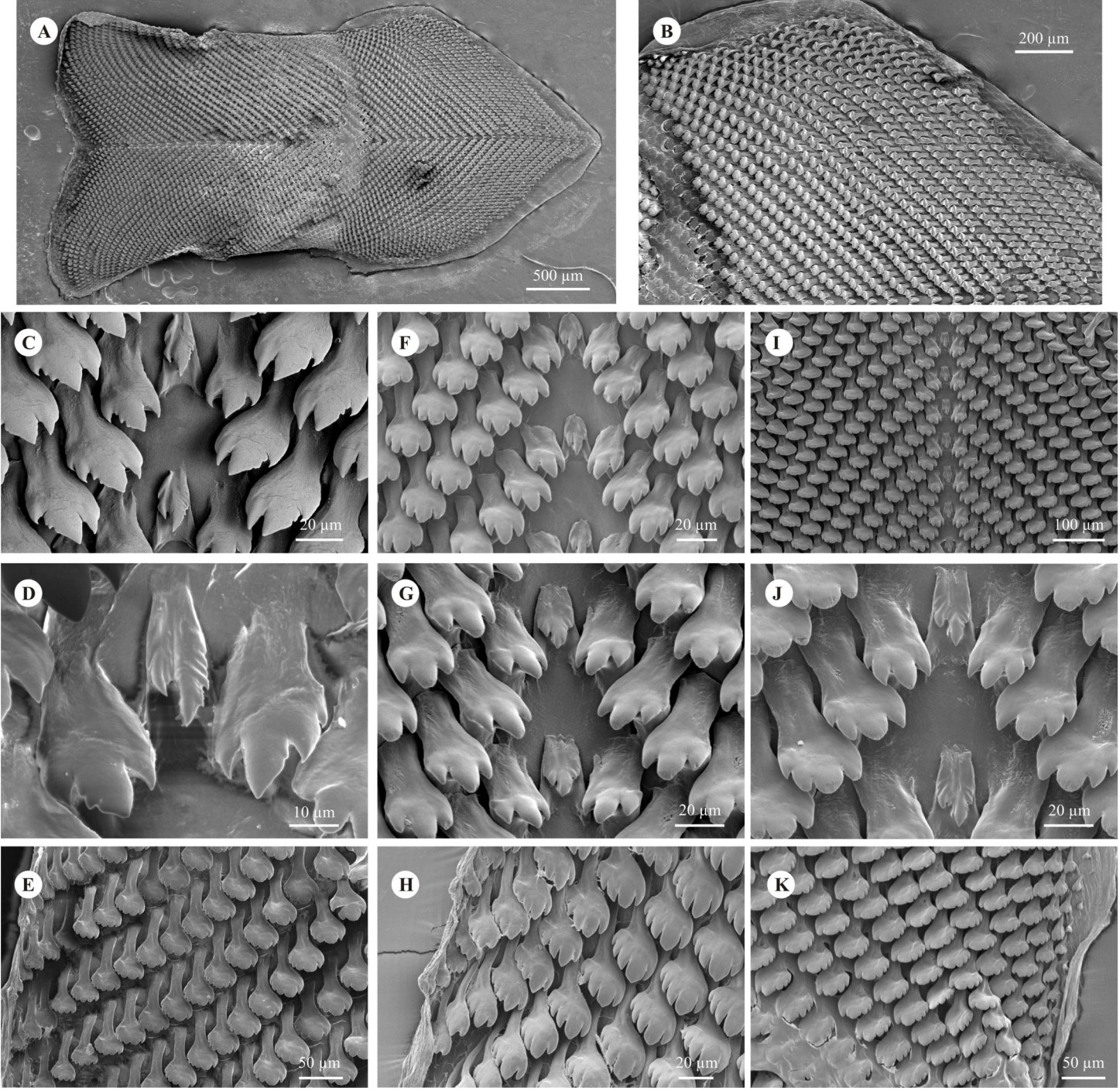

**Figure 4 Radulae.** (A–E) *Chilina nicolasi* sp. nov. from Alba Posse, Misiones province, Argentina. (F–H) *Chilina santiagoi* sp. nov. from Horacio Foerster Falls, Misiones Province, Argentina. (I–K) *Chilina luciae* sp. nov. from Pesiguero Stream, Misiones Province, Argentina. (A) General view. (B) General view of anterior part. (C, F, I) Central tooth and first lateral teeth. (D, G, J) Central tooth. (E, H, K) Lateral teeth.

surface crossed by transverse lamellae, triangular in cross section. Prepuce cylindrical, thin, with constriction marked by oblique lines arranged in a V making connection with penis sheath.

**Table 1 Average and range of five measurements for *Chilina nicolasi* sp. nov., *C. santiagoi* sp. nov., and *C. luciae* sp. nov., with specific measurements of the holotypes.**

| | | LWL | AL | TW | AW | AP |
|---|---|---|---|---|---|---|
| *Chilina nicolasi* (n = 15) | Holotype | 13.50 | 11.99 | 9.76 | 7.96 | 3.39 |
| | Mean | 13.16 | 11.74 | 9.63 | 7.45 | 3.41 |
| | SD | 1.4 | 1.16 | 1.11 | 0.88 | 0.56 |
| | Max | 16.46 | 14.26 | 12.2 | 9.49 | 4.93 |
| | Min | 10.84 | 9.69 | 8.08 | 6.34 | 2.46 |
| *Chilina santiagoi* (n = 40) | Holotype | 8.47 | 7.96 | 6.37 | 5.15 | 2.9 |
| | Mean | 7.17 | 6.77 | 5.68 | 4.56 | 2.83 |
| | SD | 1.33 | 1.29 | 0.96 | 0.79 | 0.55 |
| | Max | 9.6 | 9.04 | 7.76 | 6.08 | 4.00 |
| | Min | 4.55 | 4.3 | 3.75 | 3.00 | 1.70 |
| *Chilina luciae* (n = 10) | Holotype | 10.62 | 8.91 | 7.84 | 5.91 | 2.53 |
| | Mean | 11.54 | 9.56 | 8.51 | 6.15 | 2.78 |
| | SD | 0.86 | 0.68 | 0.71 | 0.46 | 0.25 |
| | Max | 12.91 | 10.56 | 9.82 | 7.24 | 3.09 |
| | Min | 10.54 | 8.77 | 7.65 | 5.57 | 2.38 |

Note:
LWL, last whorl length; AL, aperture length; TW, total width; AW, aperture width; AP, aperture projection.

**Radula** (Figs. 4A–4E). Average number of rows 55 (n = 3; range 52–59). Number of teeth per half row of 40–41 (n = 3). Central tooth asymmetrical, bicuspid, elongated base higher than wide, left cusp more developed. Both cusps with slight sawlike edges. Presence of marked longitudinal groove between cusps. First and second lateral teeth tricuspid or tetracuspid, with mesocone (in tricuspid, Fig. 4A) or second inner cusp (in tetracuspid, Fig. 4B) more developed and with outermost edge saw-like. Outermost teeth with thin base and five to seven cusps similarly developed. Radular formula: 40–1–40 and 41–1–41.

**Nervous system** (Fig. 3C; Table 2). All connectives between ganglia relatively thin compared to size of both ganglia and central nervous system. Left connective joining the cerebral ganglion with the pleural ganglion longer than the right one (10.1 vs 9.0% of LWL). Right pleuroparietal connective passes over the penis complex. Left pleuroparietal connective shorter than right (3.8 vs 9.2% of LWL). Parietal-subesophageal connective shorter than parietal-visceral connective (15.1 vs 23.3% of LWL). One very short connective (5.7% of LWL) linking subesophageal ganglion to visceral ganglion and closing posterior nerve ring. Pleurovisceral connectives with partial detorsion characteristic of the genus.

**Distribution** (Fig. 5). Only known from the type locality.

**DNA barcoding:** The data from the analysis of the COI of 655 bp from a paratype (MLP-Ma 14134, specimen 185) was deposited in GenBank under the number KT830419.

**Remarks:** Of the Chilinidae species for which characters of the radula have been described so far, *C. nicolasi* is the only one with the first and second lateral tooth of the outer edge of the mesocone (tricuspid) or second inner cusp (tetracuspid) serrated. The radulae of *C. gallardoi* and *C. nicolasi* have a similar number of rows and teeth per row, but

**Table 2 Ratio between the lengths of ganglia and last whorl in *Chilina nicolasi* (n = 5), *C. santiagoi* (n = 5) and *C. luciae* (n = 4).**

|  | *Chilina nicolasi* | | | *Chilina santiagoi* | | | *Chilina luciae* | | |
|---|---|---|---|---|---|---|---|---|---|
|  | Ratio | Mean | SD | Ratio | Mean | SD | Ratio | Mean | SD |
| lc–rc | 14.39 | 1.83 | 0.19 | 19.46 | 1.41 | 0.11 | 16.88 | 1.90 | 0.14 |
| lpe–rpe | 6.46 | 0.82 | 0.05 | 6.16 | 0.45 | 0.03 | 5.26 | 0.59 | 0.19 |
| lc–lpl | 10.10 | 1.28 | 0.07 | 12.20 | 0.88 | 0.17 | 7.11 | 0.80 | 0.26 |
| rc–rpl | 8.97 | 1.14 | 0.07 | 11.24 | 0.81 | 0.14 | 8.40 | 0.95 | 0.01 |
| c–p | 12.32 | 1.56 | 0.22 | 17.30 | 1.25 | 0.56 | 13.05 | 1.47 | 0.62 |
| rpl–rp | 9.22 | 1.17 | 0.29 | 12.80 | 0.93 | 0.13 | 14.01 | 1.58 | 0.31 |
| lpl–lp | 3.79 | 0.48 | 0.11 | 4.50 | 0.33 | 0.06 | 5.14 | 0.58 | 0.09 |
| lp–so | 15.08 | 1.91 | 0.18 | 19.35 | 1.40 | 0.20 | 17.25 | 1.94 | 0.38 |
| rp–v | 23.26 | 2.95 | 0.37 | 20.11 | 1.45 | 0.24 | 17.51 | 1.97 | 0.14 |
| so–v | 5.72 | 0.73 | 0.14 | 3.46 | 0.25 | – | 4.21 | 0.47 | 0.14 |

**Note:**

Abbreviations for each ganglion: c, cerebral; lc, left cerebral; lp, left parietal; lpe, left pedal; lpl, left pleural; p, pedal; rc, right cerebral; rp, right parietal; rpe, right pedal; rpl, right pleural; so, subesophageal; v, visceral. Measurements in mm.

the outermost lateral teeth in *C. nicolasi* can have up to seven cusps, while those of *C. gallardoi* have only five (Table 3). The radula of *C. iguazuensis* has more rows (57–65 vs 52–59) and teeth per half-row (43–63 vs 40–41) than *C. nicolasi* (*Gutiérrez Gregoric & Rumi, 2008*). Only three species, *C. fluminea*, *C. rushii* and *C. lilloi*, have seven cusps on the outermost lateral teeth, but the number of rows is lower (49, 48, and 44 respectively); and in *C. fluminea* and *C. lilloi* the central tooth is tricuspid (*Lanzer, 1997*; *Gutiérrez Gregoric, 2008*; *Ovando & Gutiérrez Gregoric, 2012*). The shells of *C. gallardoi* and *C. nicolasi* have two columellar teeth in the aperture, but in *C. gallardoi* both teeth are strong (as occurs in *C. fluminea* and *C. rushii*). The AL/LWL ratio in *C. gallardoi* is lower than in *C. nicolasi* (78 vs 89%: *Gutiérrez Gregoric, 2010*) while in *C. iguazuensis* it is greater (100%; *Gutiérrez Gregoric & Rumi, 2008*). In addition, *C. gallardoi* has a keel (or sub-keel) along the whorls (as does *C. rushii*), a character absent in *C. nicolasi* (and all other species).

### *Chilina santiagoi* sp. nov.

LSID urn:lsid:zoobank.org:act:4238E0F8-4452-4818-A1F2-7C1A5D8FFC4E (Figs. 2C, 2D, 4F–4H and 6).

**Type locality and type material:** Horacio Foerster Falls, Misiones Province, Argentina (27°08′S; 53°55′W), coll. D. E. Gutiérrez Gregoric, V. Núñez & R.E. Vogler, 24 March 2010.

Holotype: MLP-Ma 14135 (body in R-H, shell dry); paratypes: MLP-Ma 13417 same data (five specimens in alcohol); MLP-Ma 14136 same data (six specimens: body in R-H, shell dry).

**Other material examined:** MLP-Ma 14137 Horacio Foerster Falls, Misiones Province, Argentina (27°08′S; 53°55′W), coll. C. Galliari, May 2009 (four dry shells); MLP-Ma 14138 Moconá Falls, Misiones Province, Argentina (27°08′S; 53°53′W), coll. C. Galliari, May 2009 (12 specimens: alcohol); MLP-Ma 14139 Moconá Falls, Misiones Province,

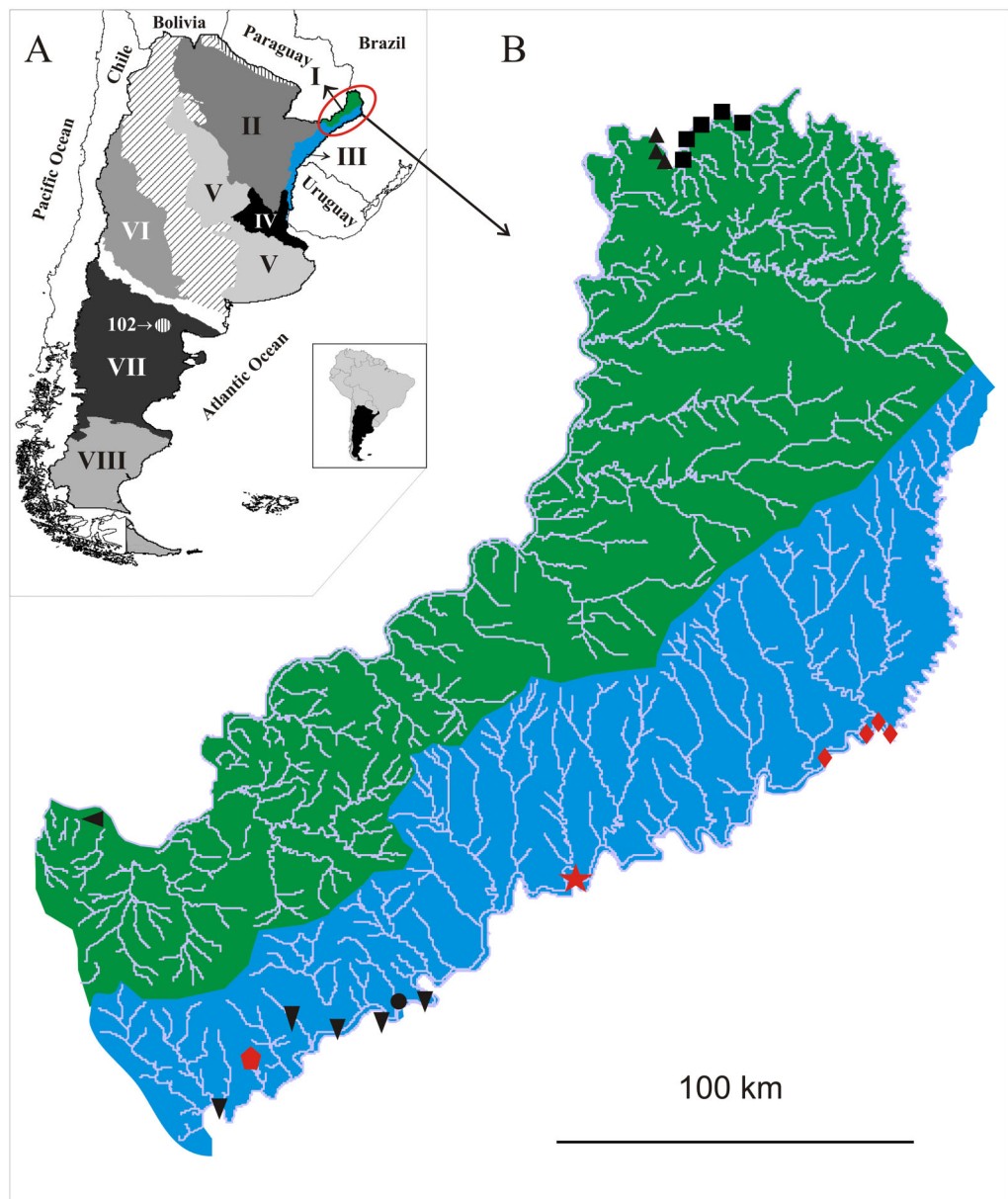

**Figure 5** **(A) Malacological provinces of Argentina, I. Misionerean; II. Middle Paraná; III. Uruguay River; IV. Lower Paraná–Río de la Plata; V. Central; VI. Cuyo; VII. Northern Patagonia; VIII. Southern Patagonia. Diagonal pattern: Transitional Zone.** (B) Species distribution of Chilinidae in the Misiones province, Argentina: green; Misionerean malacological province; light blue: Uruguay River malacological province: ♦: *Chilina santiagoi* sp. nov.; ★: *Chilina nicolasi* sp. nov.; ⬠: *Chilina luciae* sp. nov.; ▲: *Chilina megastoma*; ■: *Chilina iguazuensis*; ▼: *Chilina gallardoi*; ●: *Chilina rushii*; ◄: *Chilina guaraniana*.

Argentina (27°08′S; 53°53′W), coll. A. Rumi, S.M. Martín & I. César, October 20, 2011 (10 specimens: body in R-H, shell dry); MLP-Ma 14140 Yerba Falls, Paraíso Stream, El Soberbio, Misiones Province, Argentina (27°14′S; 54°02′W), no collector and date (two specimens: alcohol).

**Etymology:** Dedicated to the second son, Santiago, of the first author of this paper.

**Table 3 Radulae of Chilinidae species.**

| Species | Formula | NR | CT | FLT | OT |
|---|---|---|---|---|---|
| C. nicolasi | 40–1–40 or 41–1–41 | 52–59 | 2 | 3–4 | 5–7 |
| C. santiagoi | 32–1–32 or 33–1–33 | 43–44 | 2 | 3 | 5 |
| C. luciae | 40–1–40 or 41–1–41 | 50 | 2 | 4 | 5 |
| C. cuyana | 38–1–38 | 48 | 3 | 3–4 | 5 |
| C. fluminea fluminea | 30–1–30 to 34–1–34 | 49 | 3 | 3–4 | 5–7 |
| C. fluminea parva | 36–1–36 to 43–1–43 | wd | 3 | 3–4 | 4–8 |
| C. iguazuensis | 43–1–43 to 63–1–63 | 57–65 | 2 | 3 | 5 |
| C. gallardoi | 44–1–44 | 58 | 2 | 3 | 4–5 |
| C. megastoma | 42–1–42 | 49 | 2 | 3 | 4 |
| C. mendozana | 37–1–37 to 43–1–43 | 39–43 | 2 | 3–4 | 4–5 |
| C. lilloi | 39–1–39 to 43–1–43 | 42–46 | 3 | 3–4 | 5–7 |
| C. parchappii | 31–1–31 to 39–1–39 | 46–49 | 4 | 3–4 | 5 |
| C. portillensis | 38–1–38 to 41–1–41 | 50–57 | 2 | 3 | 4–5 |
| C. rushii | 35–1–35 | 48 | 2 | 3 | 5–7 |
| C. sanjuanina | 34–1–34 or 36–1–36 | 41–48 | 2 | 3 | 5 |
| C. tucumanensis | 36–1–36 to 43–1–43 | 46–58 | 3 | 3–4 | 4–5 |

Note:
NR, Number of rows; CT, number of cusps of central tooth; FLT, number of cusps of first lateral tooth; OT, number of cusps of outermost teeth; wd, without data.

**Diagnosis:** Shell small, thin, aperture projection 50% of TW; radula with asymmetrical bicuspid central tooth; penis sheath inner sculpture with regular conical pustules and longitudinal folds.

**Description:**

**Shell** (Figs. 2C and 2D). Small, thin, oval, of 3¼ whorls. Spire low and conical. Last whorl large (97% of the TL). Width 80% of LWL. Aperture expanded, 94.5% of LWL, with strong white callus. One columellar tooth. Aperture projection 50% of TW. Light brown periostracum with strong thin longitudinal reddish bands. Dimensions: see Table 1.

**Reproductive System** (Fig. 6). (i) Female reproductive system. Bursa copulatrix duct (average 4.5 mm; range 4.1–5.3; n = 4) nine times bursa sac diameter. Bursa copulatrix sac oval. Secondary bursa copulatrix long (c. 18% the length of bursa copulatrix duct), comprising of a long duct and expanded at the distal end. (ii) Male reproductive system. Muscular penis sheath, nearly twice as long as prepuce. Penis sheath inner sculpture with pustules of conical aspect and longitudinal folds. Penis slightly longer than penis sheath, robust, with outer surface cut by transverse lamellae, triangular in cross-section. Prepuce inner sculpture with numerous smooth, very tight longitudinal folds.

**Radula** (Figs. 4F–4H). Average number of rows 44 (n = 3; range 43–44). Number of teeth per half row 32–33 (n = 3). Central tooth asymmetrical, bicuspid, elongated base higher than wide, right cusp more developed and serrated, with weak longitudinal groove between the two cusps. First lateral tooth tricuspid with mesocone more developed, base of tooth same width as apical part (cusp area). Second lateral tooth tricuspid

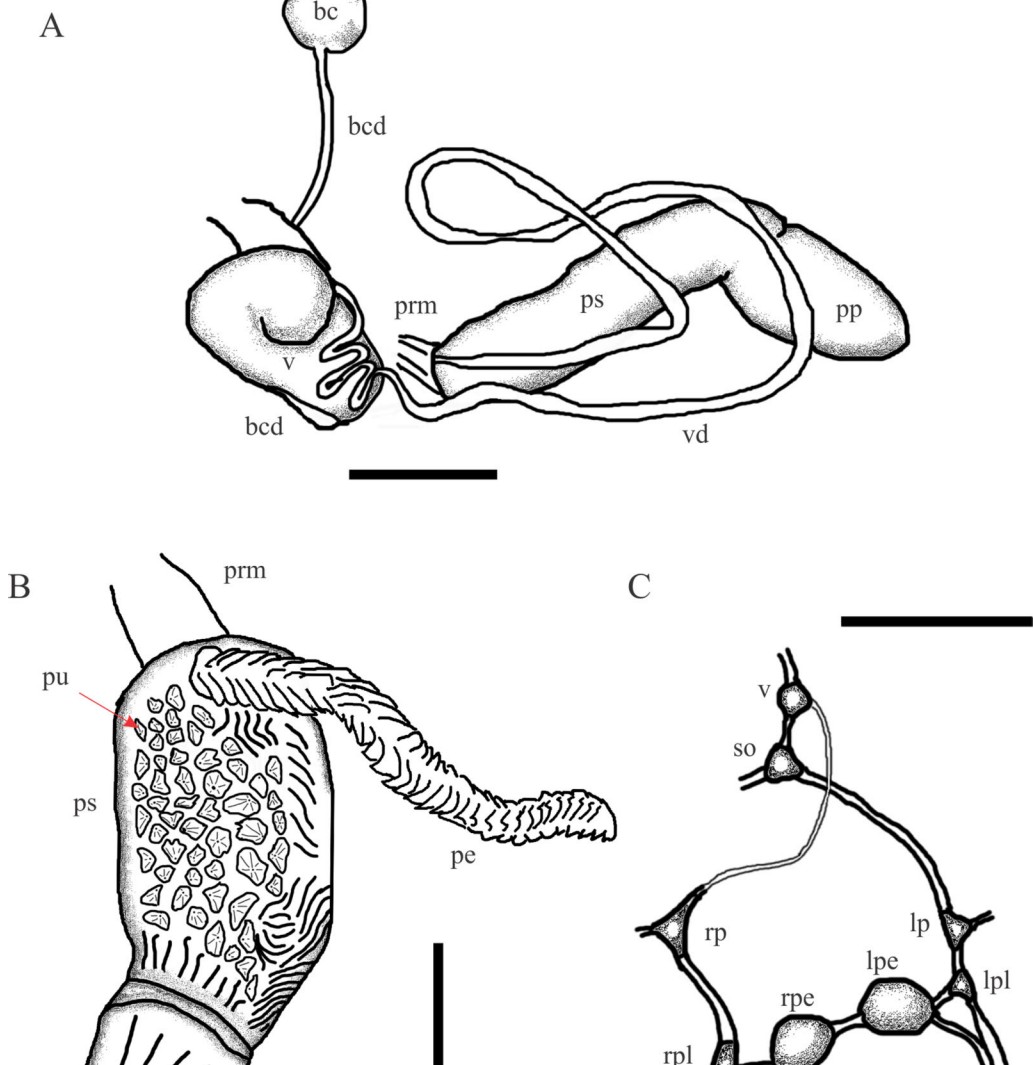

**Figure 6** *Chilina santiagoi* **sp. nov.** (A) Diagram of dorsal view of part of the reproductive system. (B) Penis inner wall. Abbreviations: ag, albumen gland; bc, bursa copulatrix; bcd, bursa copulatrix duct; pe, penis; pr, prostate; pp, preputium; prm, penis retractor muscle; ps, penis sheath; pu, pustules; v, vagina; vd, vas deferens. (C) Diagram of nervous system: Abbreviations: lc, left cerebral; lpe, left pedal; lp, left parietal; lpl, left pleural; rc, right cerebral; rpe, right pedal; rp, right parietal; rpl, right pleural; so, subesphageal; v, visceral. Scale bar: 1.0 mm.

(mainly) or tetracuspid, with mesocone (of the tricuspid) or the outermost second cusp (in the tetracuspid) more developed, base of tooth narrower than apical part of tooth. Outermost teeth with thin base, having five similarly developed cusps. Radular formula: 32–1–32 and 33–1–33.

**Nervous system** (Fig. 6; Table 2). Left connective joining the cerebral ganglion with the pleural ganglion slightly longer than the right one (12.2 vs 11.2% of LWL). Left pleuroparietal connective shorter than right one (4.5 vs 12.8% of LWL). Long connective (19.3% of LWL) linking left parietal ganglion to subesophageal ganglion, located above posterior half of columellar muscle. Long connective (20.1% of LWL) linking right parietal ganglion to visceral ganglion. One very short connective (3.5% of LWL) linking subesophageal ganglion to visceral ganglion and closing posterior nerve ring.

**Distribution** (Fig. 5). Horacio Foerster Falls is in the Yabotí Biosphere Reserve. It is a small waterfall on the Oveja Negra Stream, which flows into the Uruguay River. Water quality parameters of the Horacio Foerster Falls measured 24 March 2010, were: water temperature 23.2 °C; pH 7.62; dissolved oxygen 6.3 mg/l; conductivity 0.015 ms. Moconá Falls is in the Moconá Provincial Park, which is also in the Yabotí Biosphere Reserve. This waterfall is peculiar in the sense that it spills along a ridge parallel to the river course. Its height varies with the level of the river and it is the second largest waterfall in Misiones Province after the Iguazú Falls.

**DNA barcoding:** The data from the analysis of the COI of 655 bp and Cyt b of 388 bp from a paratype (MLP-Ma 14136, specimen 6) were deposited in GenBank under the numbers KT820416 and KT820424 respectively.

**Remarks:** The spire is not preserved in all specimens. This loss occurs in several species of Chilinidae, especially in those that inhabit fast-running water such as *C. iguazuensis* (*Gutiérrez Gregoric & Rumi, 2008*). *Chilina megastoma*, which inhabits waterfall environments, differs from *C. santiagoi* mainly in size reaching a maximum last whorl length of 9.6 mm, whereas that of *C. megastoma* is up to 17.3 mm (*Gutiérrez Gregoric, 2008*). *Chilina megastoma* has a striated shell, which *C. santiagoi* does not, and two columellar teeth, while there is one in *C. santiagoi*. Both species have thin shells. In *C. megastoma* there is a slight swelling not forming a true ganglion between the left pleural and the subesophageal ganglia (*Ituarte, 1997*; *Gutiérrez Gregoric, 2010*), but this was not detected in *C. santiagoi*. Compared with *C. nicolasi*, the shell of *C. santiagoi* is thinner, has a conical and low spire (inmersed in *C. nicolasi*), strong rather than weak bands, higher AL/LWL, AP/TW and TW/LWL ratios (94.5 vs 90%, 50 vs 34%, 80 vs 73%, respectively), and has one columellar tooth whereas there are two in *C. nicolasi*. *Chilina santiagoi* differs from *C. nicolasi* in the length of the secondary bursa copulatrix (18% of bursa copulatrix duct length vs 8%), and internal sculpture of the penis sheath (conical and longitudinal pustules vs triangular pustules). Regarding the radula, *C. santiagoi* has fewer rows of teeth (average 44 vs 55 in *C. nicolasi*) and fewer teeth per half row (average 32 vs 40); the developed cusp of the central tooth (in both the tooth is bicuspid) is the right cusp (left in *C. nicolasi*) and the cusp has only one serrated edge (both in *C. nicolasi*), and the outermost lateral teeth have five cusps (up to seven in *C. nicolasi*).

### *Chilina luciae* sp. nov.

LSID urn:lsid:zoobank.org:act:FE46F318-BA47-4D5B-8AD2-266C63EB87A4 (Figs. 2E, 2F, 4I–4K and 7).

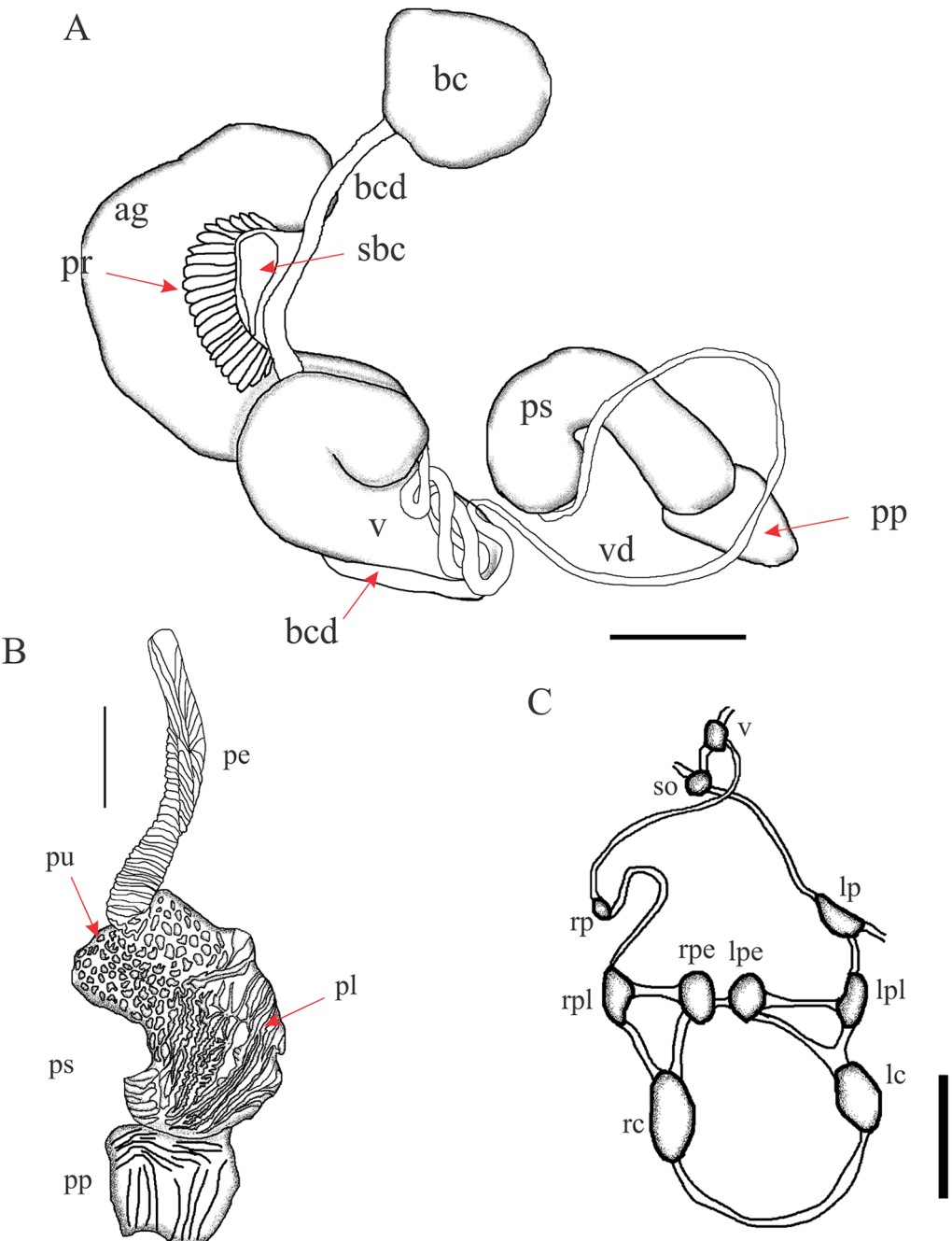

**Figure 7** *Chilina luciae* **sp. nov.** (A) Diagram of dorsal view of part of the reproductive system. (B) Penis inner wall. Abbreviations: ag, albumen gland; bc, bursa copulatrix; bcd, bursa copulatrix duct; lf: longitudinal folds; pe, penis; pr, prostate; pp, preputium; ps, penis sheath; pu, pustules; sbc, secondary bursa copulatrix; v, vagina; vd vas deferens. (C) Diagram of nervous system: Abbreviations: lc, left cerebral; lpe, left pedal; lp, left parietal; lpl, left pleural; rc, right cerebral; rpe, right pedal; rp, right parietal; rpl, right pleural; so, subesphageal; v, visceral. Scale bar: 1.0 mm.

**Type locality and type material:** Pesiguero stream, Misiones Province, Argentina (27°58′S; 55°26′W), coll. D. E. Gutiérrez Gregoric, March 21, 2010.

Holotype: MLP-Ma 14141 (alcohol); Paratypes: MLP-Ma 13413 same data (5 specimens: foot in alcohol, body R-H, shell dry); MLP-Ma 14142 same data (4 specimens: alcohol).

**Etymology:** Dedicated to the daughter, Lucía, of the first author of this paper.

**Diagnosis:** Shell thick, aperture projection 33% of TW; radula, central tooth bicuspid and with saw-like external side, first lateral tetracuspid; prepuce 37% of length of penis sheath; penis sheath inner sculpture with two regions, one with polygonal pustules and the other with longitudinal zigzag folds.

**Description:**

**Shell** (Figs. 2E and 2F). Thick, slightly elongated. Spire eroded. Width 74% of LWL. Aperture somewhat expanded, of 83% of LWL, with strong white callus. Two strong columellar teeth. Aperture projection 33% of TW. Light reddish periostracum with some dark brown spots. Dimensions: see Table 1.

**Reproductive System** (Fig. 7). (i) Female reproductive system. Bursa copulatrix duct (average 4.7 mm; range 4.5–4.8; n = 3) four times bursa sac diameter. Bursa copulatrix sac spherical. Secondary bursa copulatrix short (c. 11% of the length of bursa copulatrix duct), cylindrical, expanded at its distal portion. (ii) Male reproductive system. Muscular penis sheath, a little more than twice the length of prepuce (2.1 vs 0.8 mm). Penis sheath inner sculpture with polygonal pustules and longitudinal zigzag folds. Penis 92% the length of penis sheath, robust, with outer surface crossed by transverse lamellae, triangular in cross section. Inner sculpture of prepuce with numerous smooth, very tight longitudinal folds.

**Radula** (Figs. 4I–4K). Number of rows 50 (n = 2). Number of teeth per half row 40–41 (n = 2). Central tooth asymmetrical, bicuspid, elongated base higher than wide, both cusps with serrated edges. First lateral tooth tetracuspid with innermost second cusp more developed, base of tooth same width as apical part (cusp area). Second lateral tooth tetracuspid, with innermost second cusp more developed, base of tooth narrower than apical part of tooth. Outermost teeth with thin base, having five similarly developed cusps. Radular formula: 40–1–40 and 41–1–41.

**Nervous System** (Fig. 7; Table 2). Left connective joining the cerebral ganglion with the pleural ganglion longer than the right one (10.3 vs 8.8% of LWL). Left pleuroparietal connective smaller than the right one (5.9 vs 17.7% of LWL). Long connective (22.1% of LWL) linking left parietal ganglion to subesophageal ganglion, located above posterior half of columellar muscle. Long connective (16.2% of LWL) linking right parietal ganglion to visceral ganglion. One very short connective (3.5% of LWL) linking subesophageal ganglion to visceral ganglion and closing posterior nerve ring.

**Distribution** (Fig. 5). Only known from the type locality. Pesiguero Stream drains into the Uruguay River and is in the Concepción de la Sierra District of Misiones Province. The Uruguay River is 10 km from the collection site.

**DNA barcoding:** The data from the analysis of the COI of 655 bp and Cyt b of 388 bp from a paratype (MLP-Ma 14142, specimen 186) were deposited in GenBank under the numbers KT820420 and KT820425 respectively.

**Remarks:** *Chilina luciae*, like *C. gallardoi* and *C. rushii*, was recorded in the rapids of a stream that flows into the Uruguay River. *Chilina luciae* differs from both those species by not having a shell keel. *Chilina luciae* has two strong columellar teeth as in *C. gallardoi*, *C. fluminea* and *C. rushii*; while *C. nicolasi* also has two columellar teeth, but weak ones. The AL/LWL ratio in *C. luciae* is lower than in *C. nicolasi* and *C. santiagoi* (83 vs 90 and 95% respectively), but higher than in *C. gallardoi* (78%) (*Gutiérrez Gregoric, 2010*). The AP/TW and TW/LWL ratios in *C. luciae* and *C. nicolasi* are similar (33 vs 35% and 74 vs 73%, respectively) and all lower than in *C. santiagoi* (50 and 80% respectively). *Chilina luciae* differs from *C. nicolasi* and *C. santiagoi* in the length of the secondary bursa copulatrix (11% of bursa copulatrix duct length vs 8 and 18% respectively), and internal sculpture of the penis sheath (polygonal and zigzag longitudinal pustules vs triangular pustules and conical and longitudinal pustules, respectively. The radula of *C. luciae* has similarities and differences with other species, but in no case is equal to any of them (Table 3; *Gutiérrez Gregoric, 2010*; *Ovando & Gutiérrez Gregoric, 2012*; *Gutiérrez Gregoric, Ciocco & Rumi, 2014*). The first lateral tooth of *C. luciae* is tetracuspid like in *C. fluminea* (*Gutiérrez Gregoric, 2010*).

## Molecular analyses

Four novel sequences of 388 bp for Cyt b (*C. nicolasi*, 1; *C. luciae*, 1; *C. fluminea*, 1; *C. gallardoi*, 1) and 15 sequences of 655 bp for COI (*C. nicolasi*, 1; *C. santiagoi*, 1; *C. luciae*, 1; *C. fluminea*, 5; *C. rushii*, 1; *C. gallardoi*, 1; *C. iguazuensis*, 4; *C. megastoma*, 1) were obtained (Table 4). BLAST searches identified Cyt b and COI sequences as similar to other Hygrophila, ruling about possible contamination with DNA from other sources.

The COI sequences obtained here for *Chilina nicolasi* and *C. santiagoi* differ by *c.* 1.2%, while those of *C. luciae* differ from the other two species described in this work by 3.8% (Table 5). The two phylogenetic analyses (Fig. 8) recovered two well-supported sister clades with high posterior probabilities and bootstrap values. Both analyses showed two groups within the Chilinidae species, one belonging to the Misionerean malacological province and the other representatives from the other two malacological provinces (Uruguay River and Lower Paraná-Río de la Plata).

The Cyt b sequences obtained here for *Chilina santiagoi* and *C. luciae* differ by *c.* 4%, a distance similar to that found in COI, again indicating that *C. santiagoi* is a new species. Distances of both new species from the other two species from which this gene was sequenced are similar (Table 6).

## DISCUSSION

This report provides anatomical, molecular-genetic, and distributional information on the species of *Chilina* of lotic environments from the Uruguay River malacological province, increasing the number of known freshwater gastropod species in this province from 51 to 54. This province exhibits the highest freshwater gastropod richness in Argentina, and contains the highest number of vulnerable (14) and endemic species (15) (*Núñez, Gutiérrez Gregoric & Rumi, 2010*). These new endemic species from the

**Table 4 Information on the specimens of *Chilina* used in DNA sequence analysis, with a lymnaeid for comparison.**

| Species | Site/Malacological province | GenBank COI | Cyt b |
|---|---|---|---|
| *Chilina nicolasi* | Alba Posse/III | KT830419* | – |
| *Chilina santiagoi* | H. Foerster Falls/III | KT820418* | KT820424* |
| *Chilina luciae* | Pesiguero Stream/III | KT820420* | KT820425* |
| *Chilina gallardoi* | Monte Caseros/III | KT820421* | KT820427* |
| *Chilina rushii* | Gualeguaychú river/III | KT820423* | – |
| *Chilina fluminea* | Punta Lara/IV | KT807833*# | – |
| *Chilina fluminea* | Punta Lara/IV | KT807832*# | – |
| *Chilina fluminea* | Punta Lara/IV | KT807831*# | – |
| *Chilina fluminea* | Punta Lara/IV | KT807834*# | – |
| *Chilina fluminea* | Punta Lara/IV | KT820422* | KT820426* |
| *Chilina iguazuensis* | Iguazú National Park/I | KT807837*# | – |
| *Chilina iguazuensis* | Iguazú National Park/I | KT807838*# | – |
| *Chilina iguazuensis* | Iguazú National Park/I | KT807836*# | – |
| *Chilina iguazuensis* | Iguazú National Park/I | KT807835*# | – |
| *Chilina megastoma* | Iguazú National Park/I | KT807839*# | – |
| *Chilina sanjuanina* | Aguas Negras/VI | KC347574 | – |
| *Chilina mendozana* | Uspallata/VI | KC347575 | – |
| *Lymnaea diaphana* | | JF909501 | – |

**Notes:**
* New sequences.
# Sequences generated by the BOLD program. Numerals correspond to the malacological provinces: I Misionerean, III Uruguay River, IV Lower Paraná–Río de la Plata, VI Cuyo.

**Table 5 Pairwise genetic divergence (Kimura two-parameter, %) among species of *Chilina* assessed by means of COI gene sequences.**

| | | 1 | 2 | 3 | 4 | 5 | 6 | 7 | 8 | 9 | 10 | 11 | 12 | 13 |
|---|---|---|---|---|---|---|---|---|---|---|---|---|---|---|
| 1 | *C. fluminea* (KT807831/33; KT820422) | | | | | | | | | | | | | |
| 2 | *C. fluminea* (KT807832) | 0.15 | | | | | | | | | | | | |
| 3 | *C. fluminea* (KT807834) | 0.31 | 0.46 | | | | | | | | | | | |
| 4 | *C. rushii* (KT820423) | 1.24 | 1.39 | 1.55 | | | | | | | | | | |
| 5 | *C. gallardoi* (KT820421) | 2.98 | 3.14 | 3.3 | 2.66 | | | | | | | | | |
| 6 | **C. santiagoi** sp. nov. (KT820418) | 2.97 | 3.13 | 3.29 | 3.14 | 2.66 | | | | | | | | |
| 7 | **C. nicolasi** sp. nov. (KT820419) | 2.97 | 3.13 | 3.29 | 3.14 | 2.34 | 1.24 | | | | | | | |
| 8 | **C. luciae** sp. nov. (KT820420) | 5.27 | 5.44 | 5.61 | 5.45 | 4.29 | 3.79 | 3.79 | | | | | | |
| 9 | *C. iguazuensis* (KT807838) | 8.01 | 8.19 | 7.66 | 8.75 | 7.68 | 6.67 | 7.35 | 8.58 | | | | | |
| 10 | *C. iguazuensis* (KT807833/37) | 7.84 | 8.01 | 7.48 | 8.57 | 7.5 | 6.49 | 7.17 | 8.4 | 0.15 | | | | |
| 11 | *C. iguazuensis* (KT807836) | 7.49 | 7.67 | 7.14 | 8.22 | 7.16 | 6.48 | 6.83 | 8.05 | 0.46 | 0.31 | | | |
| 12 | *C. megastoma* (KT807839) | 8.52 | 8.7 | 8.17 | 8.9 | 9.63 | 8.72 | 8.9 | 9.62 | 7.83 | 7.65 | 7.66 | | |
| 13 | *C. sanjuanina* (KC347575) | 11.8 | 12 | 11.4 | 12 | 11.6 | 11.3 | 11.8 | 12 | 10.7 | 10.5 | 10.1 | 13.8 | |
| 14 | *C. mendozana* (KC347574) | 10.8 | 11 | 10.5 | 10.3 | 11.4 | 10.7 | 11.2 | 10.7 | 11.2 | 11 | 10.7 | 12.8 | 3.64 |

**Note:**
GenBank accession numbers are indicated in parentheses.

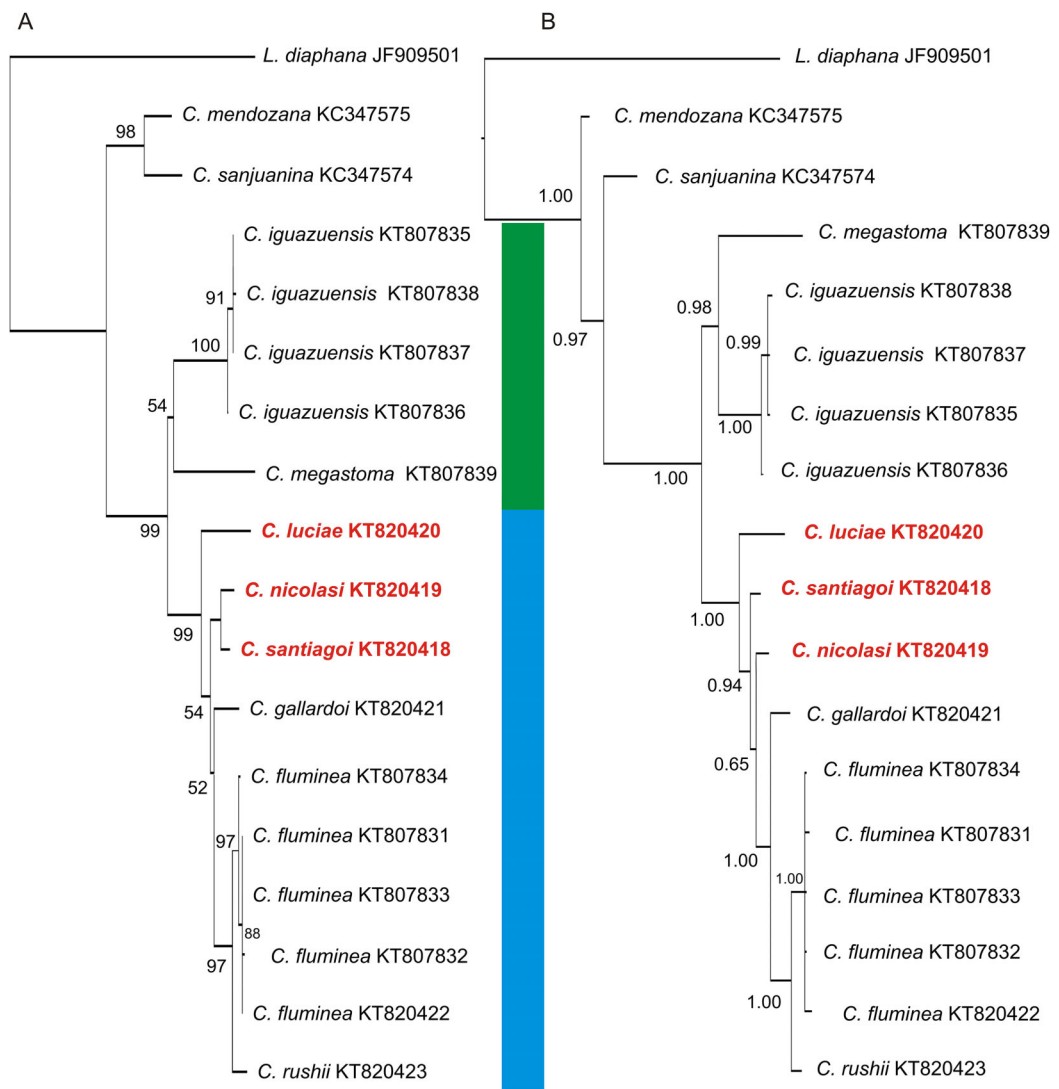

**Figure 8 Phylogenetic trees of Chilinidae from the Del Plata basin based on a 655-bp fragment of the COI gene.** (A) NJ tree. (B) Bayesian consensus tree. The support values, bootstrap values NJ and posterior probabilities (Bayesian inference), are shown above and below the branches. The trees contains two well supported clades corresponding to the species of Misionerean (green bar) and Uruguay River and Lower Paraná–Río de la Plata (light blue bar) malacological provinces. The numbers within the clades are the corresponding GenBank accession numbers.

**Table 6 Pairwise genetic divergence (Kimura two-parameter, %) among species of *Chilina* assessed by means of Cyt b gene sequences.**

|   |   | 1 | 2 | 3 |
|---|---|---|---|---|
| 1 | *C. fluminea* (KT820426) |   |   |   |
| 2 | ***C. santiagoi* sp. nov.** (KT820424) | 2.91 |   |   |
| 3 | ***C. luciae* sp. nov.** (KT820425) | 4.26 | 3.99 |   |
| 4 | *C. gallardoi* (KT820427) | 4.00 | 3.46 | 4.82 |

**Note:**
GenBank accession numbers are indicated in parentheses.

Uruguay River add further support to the suggestion that this river is a diversity hotspot of freshwater gastropods (*Strong et al., 2008*). The family Chilinidae is now represented by 24 species in Argentina, of which 17 are endemic.

From an anatomical viewpoint, the new species exhibit differences in shell, radula and reproductive system characteristic, especially in the sculpture of the penis sheath. In species recently described and re-described by *Ovando & Gutiérrez Gregoric (2012)* and *Gutiérrez Gregoric, Ciocco & Rumi (2014)* differences in the above characters were also found.

The interspecific genetic distances found in the present study for COI were 1.2% or greater, and the intraspecific distances lower than 0.5%. Studies in Lymnaeidae (Gastropoda, Hygrophila) have suggested a similar interspecific genetic distance for COI among neotropical species (*Correa et al., 2011*). For land molluscs, *Davison, Blackie & Scothern (2009)* estimated interspecific genetic distances of 12% and intraspecific of 3%, but noted that the interspecific genetic distances can also be quite low, around 1%. For this reason, we suggest that an integrative vision is necessary—one that complements conchological and anatomical information with molecular genetics and ecological data.

The phylogenetic analyses of Chilinidae confirmed segregation of the freshwater gastropod fauna of Misiones Province from those of other provinces, as suggested by *Núñez, Gutiérrez Gregoric & Rumi (2010)*. The species described here from the Uruguay River malacological province are distinct from those of the Misionerean malacological province, *e.g. C. megastoma* and *C. iguazuensis*. Nevertheless, the species from the Río de la Plata River (*C. fluminea* and *C. rushii*) are more closely associated with those of the Uruguay River. The species in the Cuyo malacological province (*C. mendozana* and *C. sanjuanina*) are distinct from those from the Del Plata basin. Likewise, species of *Aylacostoma* (Thiaridae) and *Acrorbis* (Planorbidae) in the Misioneran malacological province have not been recorded in the Uruguay River malacological province (*Núñez, Gutiérrez Gregoric & Rumi, 2010*). Despite malacological differences, ichthyological classifications (*Ringuelet, 1975*; *López, Morgan & Montenegro, 2002*) suggest that Misiones Province (as a political division) should be considered in its entirety as an ecoregion.

With the examples described here, the number of endemic species known in waterfall environments increases. Thus, species living in Misiones Province are *Chilina megastoma*, endemic to Iguazú National Park, *Acrorbis petricola*, from the waterfalls of Iguazú National Park and the Encantado Falls (Aristólubo del Valle), and *Chilina santiagoi* in the Uruguay River. In addition, *C. nicolasi* and *C. luciae* have been added to the species recorded in rapids along rivers in Misiones Province, which include *C. iguazuensis*, *Sineancylus rosanae*, *Felipponea* spp. and *Aylacostoma* spp. A hydroelectric dam is going to be built in the area where *C. nicolasi* were collected. This hydroelectric dam will raise the level the Uruguay River, causing the disappearance of the environment inhabited by the species. These endemic species should be taken into account by government agencies before the construction of dams that modify these types of environments in the Uruguay River.

## ACKNOWLEDGEMENTS

We thank the curator and technical staff of the malacological collection of La Plata Museum (MLP) G. Darrigran and M. Tassara for their generosity in lending the material under study, and R. Vogler, V. Núñez, and A. Rumi for support during field work. We are especially grateful to Dr. M. Griffin, Dr. R. Cowie and Dr. C. Lydeard for her helpful comments. Dr. D.F. Haggerty, a retired academic career investigator and native English speaker, edited the final draft of the manuscript.

### Funding

The authors received no funding for this work.

### Competing Interests

The authors declare that they have no competing interests.

### Author Contributions

- Diego E. Gutiérrez Gregoric conceived and designed the experiments, performed the experiments, analyzed the data, contributed reagents/materials/analysis tools, wrote the paper, prepared figures and/or tables, reviewed drafts of the paper.
- Micaela de Lucía analyzed the data, contributed reagents/materials/analysis tools, wrote the paper, prepared figures and/or tables, reviewed drafts of the paper.

### Field Study Permissions

The following information was supplied relating to field study approvals (i.e., approving body and any reference numbers):

Ministerio de Ecología, Recursos Naturales Renovables y Turismo Dirección de Biodiversidad, Gobierno de la Provincia de Misiones authorized this research by letter.

### DNA Deposition

The following information was supplied regarding the deposition of DNA sequences:

GenBank accession numbers:

KT807831, KT807832, KT807833, KT807834, KT807835, KT807836, KT807837, KT807838, KT807839, KT820418, KT820419, KT820420, KT820421, KT820422, KT820423, KT820424, KT820425, KT820426, KT820427.

### Data Deposition

Museo de La Plata, Argentina (MLP-Ma): http://www.museo.fcnym.unlp.edu.ar/zoologia_invertebrados_colecciones_seccion_malacologia.

### New Species Registration

The following information was supplied regarding the registration of a newly described species:

Publication LSID: urn:lsid:zoobank.org:pub:3140E36D-B1F5-4C1B-9F3C-0081CDE88B00.

Chilina luciae Gutiérrez Gregoric LSID urn:lsid:zoobank.org:act:FE46F318-BA47-4D5B-8AD2-266C63EB87A4.

Chilina nicolasi Gutiérrez Gregoric LSID urn:lsid:zoobank.org:act:A7D18E3D-1CA1-470F-A5B6-3EB070C994C3.

Chilina santiagoi Gutiérrez Gregoric LSID urn:lsid:zoobank.org:act:4238E0F8-4452-4818-A1F2-7C1A5D8FFC4E.

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
