# Peer review of "Freshwater gastropods diversity hotspots: three new species from the Uruguay River (South America)"

_PeerJ, doi:10.7717/peerj.2138_

## Round 0.1 · original submission · Major Revisions

I am sorry for the delay in getting this back to you, but we struggled to get referees as we approached the Holiday season.

In the end, one referee has fairly extensive comments and suggestions for improvements to the manuscript, whereas the other provides little in the way of constructive feedback. Rather than delay your submission further, I will make a decision based on these. In reading through the manuscript, I find that I am more in agreement with the referee who makes the detailed comments on your manuscript and has included a marked-up manuscript for suggested revisions to the text. I note that their comments on revision to the Code are very well taken, and in order for the species as described to be valid, I agree that the authors need to follow the provided link and ensure that all requirements for electronic publication are met (and detail exactly how for the referees) as suggested by the reviewer. Finally, as the referee points out the molecular phylogenetic analysis is simply not up to the current standard in the field. In fact, some argue that the age of such analyses is now past (http://www.sciencedirect.com/science/article/pii/S0169534715001093 ). I do not agree that we need whole genomes to address such questions, but some relatively recent examples (such as http://journals.plos.org/plosone/article?id=10.1371/journal.pone.0046105 ) should help you to see how others have handled the issue of taxon sampling and phylogenetic approaches to similar studies in other taxa; please feel free to contact me for additional guidance if this example is not sufficient.

·

Basic reporting

As far as I can tell it adheres to all PeerJ policies.

The article is written in English and has been reviewed by a native English speaker. However, in many places it still requires minor editing and in other places the meaning is unclear . I have indicated such instances on the marked up manuscript that I am submitting with my review.

The introduction/background is sufficient and appropriate literature is referenced.

The structure differs from the standard as this, unusually for PeerJ, is a taxonomic paper describing new species. It therefore has a "Systematics" section, which is perfectly acceptable and the norm in a taxonomic paper.

Figures are relevant and appropriately described and labeled. I assume they are of sufficient resolution - at least I can see the detail I need to see. However, I do think the reproductive system illustrations are rather lacking in detail by modern standards. I also would like to have seen SEMs of the full width of the radulae.

The submission is self-contained and represents an appropriate unit of publication and includes all the results. There is no hypothesis as this is taxonomy not experimental science. I do not see it as a part of an unjustifiably subdivided work.

As far as I can tell, all appropriate raw data have been made available, although I suppose it could be argued that giving individual values as well as means, SDs and ranges would be better, but not absolutely necessary, in my view (but see my comments on the manuscript).

Experimental design

The submission is within the scope of the journal.

This is a taxonomic paper describing three new species. It is not an experimental paper and does not pose a question or set up a hypothesis, which is not the nature of taxonomic research. However, describing three new species obviously fills a knowledge gap.

The research is decent but as presented is not of the highest standard for a modern taxonomic paper. See my comments on the marked up manuscript.

The research seems to have been conducted ethically.

Validity of the findings

The data are robust; what statistics are used (purely descriptive means SDs and ranges) are okay but probably only ranges are necessary; this is not an experimental study so no controls are involved.

The data are provided in the paper or on GenBank.

The conclusions are appropriately stated, fulfill the goal of the paper and go no further beyond that. However, I have some reservations about the validity of at least one of the new species as I question the distinctness of C. nicolasi and C. santiagoi - see comments on the marked up manuscript.

I make no judgement on impact etc. There is no pointless repetition of results.

All my comments of detail are on the marked up manuscript.

Comments for the author

I have marked up your manuscript with many comments, questions, corrections and improvements, both regarding the English and the science.

I am not expert in molecular phylogenetics but there are preferable ways to build a phylogeny than neighbor-joining I also note that you did not include ALL Chilina species in your phylogeny – how did you choose which to include and could that impact your results and interpretations? I hope the handling editor can advise you on these issues, as he is more expert than I am in this area.

This is the first time I have seen a paper that publishes new species descriptions following the amendment to the Code regarding electronic publication. I therefore ask that you look at the amendment extremely carefully in order to make sure that every requirement is fulfilled, and explain how each requirement is fulfilled, in sequence as in the amendment. The amendment must also be referenced. I have provided a link to the amendment in the marked up manuscript. You do not want your paper and descriptions to be "unavailable" according to the Code because of a technicality. This is extremely important.

Finally, I think you need to explain more clearly your approach to determining that these are new species. Did you build a phylogeny from your specimens, then identify monophyletic lineages from that phylogeny and then try to put existing names on those lineages, find that there was no name for some of them, and therefore described those lineages as new species? Or did you use specimens that you previously identified morphologically as known species, with additional specimens that you could not name and that you therefore determined morphologically as belonging to three new species, and then built the phylogeny? The reason I think you need this explanation is because I am not convinced of the real distinctness of C. nicolasi and C. santiagoi.

·

Basic reporting

The paper is a very well written account of the diversity hotspots of freshwater gastropods from the Uruguay River (South America) with a detailed description of 3 new species using both molecular and morphological characters.

Experimental design

The design is more than adequate. The authors examine shell, soft anatomy, radula, DNA sequences of two mitochondrial genes to distinguish the three new species and conduct a phylogenetic analysis to examine phylogenetic context of three new species within the family.

Validity of the findings

The three species possess distinguishing characters that indicate they are indeed new species and phylogenetic analysis reveals distinction between different river systems. The paper also is important from a conservation perspective due to the fact the species will be negatively impacted should the river system be impounded with a loss of habitat.

Comments for the author

The authors should be commended for conducting a rigorous study to delineate and describe three new gastropod species in an aquatic hotspot of South America.

---

## Round 0.2 · Minor Revisions

Overall the manuscript is much improved, and my impression is that the manuscript will be acceptable for publication after the English grammar and spelling are corrected. The referee has taken the time and effort to offer extensive suggestions on improving the English usage in the annotated manuscript attached, and we all owe them a debt for the investment in that effort. I ask that you make these changes and carefully proofread the manuscript before resubmission. I will complete a final edit and review of the text to ensure that the text is ready for publication at that time.

·

Basic reporting

The manuscript is much improved from the original version. However, it still needed a lot of editing, especially new text. All my comments are on the attached pdf, derived from a Word file - I can send the Word file if the editor wishes, as it will facilitate making the numerous editorial corrections.

I still think a diagram of the shell measurements is essential, especially as the reference now provided explaining what these are is not readily available.

I still do not like their interpretation of the central tooth cusps, but perhaps it's a matter of pinion and I will let it go.

How are the holotypes stored - are they shells only?

Experimental design

No additional comments

Validity of the findings

No additional comments

Comments for the author

No additional comments

---

## Round 0.3 · accepted · Accept

I am sorry for the delay in getting this back to you. I read your revision and your rebuttal letter and felt that you had addressed the comments of the referees sufficiently in the revision with the exception of one: [U37] in which the referee asks about the reliability of species separated on 1.24% divergence. You responded that "We consider genetic information as a character, which may or may not differ." I have discussed this issue at some length with several colleagues (hence the delay) and all agreed that there must be underlying genetic differences to describe a species. A given locus (such as COI) may not differ greatly among species, but clearly the morphological traits on which the description is based must have an underlying genetic basis, or the traits would be plastic and describe ecomorphs rather than species. Each of the experts with whom I spoke felt that such statements are indefensible in this age of genomic information, but in re-reading this section several times in the course of my discussions, I find myself in agreement with you that any specific cutoff value of divergence is arbitrary and the real metric by which I consider a species to be valid is the mean intraspecific variation in comparison to the mean interspecific variation among individuals that are grouped by diagnostic morphologies. Several I spoke to argued that it would be possible to get such morphologically discrete groups by selective sampling of the ends of a continuum and without a population genetic sample size to look at genetic diversity, the divergence within as opposed to among these groups remained uncertain. However, you seem to have that information based on what you present in lines 470-471: The interspecific genetic distances found in the present study for COI were above 1.2% or greater, and the intraspecific distances lower than 0.5%. I believe it would be more convincing to the average reader if you were to present these data in a plot of the pairwise intraspecific and pairwise interspecific distances among your samples to show that they are non-overlapping (if I understand this sentence correctly). I leave it to you whether or not you wish to include such a plot in your manuscript or as a supplement, but it seems to me in my discussions with colleagues that would be a powerful addition to convince skeptics.

In the end, I find myself in agreement with your argument that "an integrative vision is necessary—one that complements conchological and anatomical information with molecular genetics, and ecological data" at least when we are discussing a couple of small mtDNA loci of unknown taxonomic reliability. As such, I have decided to accept your manuscript, and apologize for taking far longer to evaluate your manuscript than I intended, but also felt that you deserve careful consideration of your work and the referee comments rather than a hasty decision. I suppose we will see if future genomic work overturns any of the taxonomic distinctions you raise here, but I do not see any value in preventing publication of your work on unproven suspicions, especially when there is concern over the conservation status of these species. Therefore, I have decided to move your manuscript forward into production.